# Role of Plant-Growth-Promoting Rhizobacteria in Plant Machinery for Soil Heavy Metal Detoxification

**DOI:** 10.3390/microorganisms12040700

**Published:** 2024-03-29

**Authors:** Haichen Qin, Zixiao Wang, Wenya Sha, Shuhong Song, Fenju Qin, Wenchao Zhang

**Affiliations:** School of Chemistry and Life Sciences, Suzhou University of Science and Technology, Suzhou 215009, China

**Keywords:** heavy metal, plant-growth-promoting rhizobacteria (PGPR), soil contamination, bioremediation, heavy metal detoxification

## Abstract

Heavy metals migrate easily and are difficult to degrade in the soil environment, which causes serious harm to the ecological environment and human health. Thus, soil heavy metal pollution has become one of the main environmental issues of global concern. Plant-growth-promoting rhizobacteria (PGPR) is a kind of microorganism that grows around the rhizosphere and can promote plant growth and increase crop yield. PGPR can change the bioavailability of heavy metals in the rhizosphere microenvironment, increase heavy metal uptake by phytoremediation plants, and enhance the phytoremediation efficiency of heavy-metal-contaminated soils. In recent years, the number of studies on the phytoremediation efficiency of heavy-metal-contaminated soil enhanced by PGPR has increased rapidly. This paper systematically reviews the mechanisms of PGPR that promote plant growth (including nitrogen fixation, phosphorus solubilization, potassium solubilization, iron solubilization, and plant hormone secretion) and the mechanisms of PGPR that enhance plant–heavy metal interactions (including chelation, the induction of systemic resistance, and the improvement of bioavailability). Future research on PGPR should address the challenges in heavy metal removal by PGPR-assisted phytoremediation.

## 1. Introduction

Since the 20th century, amidst the rapid development of the social economy, humans have wantonly burned fossil fuels, mined gold deposits, and used pesticides, fertilizers, and other chemicals, causing a significant rise in heavy metal pollution in the soil environment, both in terms of intensity and geographical extent [1]. According to the China National Soil Pollution Survey Bulletin, the heavy metals of Cd, Hg, As, Cu, Pb, Cr, Zn, and Ni in soil exceeded the acceptable standard by 7.0%, 1.6%, 2.7%, 2.1%, 1.5%, 1.1%, 0.9%, and 4.8%, respectively. They enter the soil environment through fertilization, irrigation, atmospheric settlement, and other means, causing soil and groundwater pollution [2]. Soil contaminated by heavy metals shows ecological imbalance, biodiversity loss, soil erosion, desertification, acidification, salinization, declining soil fertility, soil consolidation and soil subsidence [3,4]. Moreover, heavy metals can only be transferred from one chemical state to another due to non-biodegradability [5]. Because of their remarkable environmental persistence, heavy metals in soil will be absorbed and accumulate in plants, then pose hazards to humans and animals via the food chain [6]. Hence, it is crucial to remediate these toxic heavy metals in soil.

Although traditional physicochemical treatments such as soil replacement, soil washing, and chemical solidification have partly reduced the migration of soil heavy metals to groundwater, their applications are limited due to high energy consumption, secondary pollution, and the breakdown of soil aggregate structures [7]. In contrast, bioremediation (such as biosorption or bioaccumulation) is a simple, environmentally friendly and sustainable method that has raised significant interest [8]. Bioremediation relies on natural biological processes to eliminate toxic pollutants by means of microorganisms, green plants, or their enzymes. Among the various bioremediation methods, phytoremediation is widely accepted for its economic feasibility and eco-friendliness. However, low biomass yield, sensitivity to polymetallic substances, and shallow root systems limit the efficiency of phytoremediation. A promising method to improve the repair efficiency of plants with high metal concentrations is the employment of plant-growth-promoting rhizobacteria (PGPR) as a tool to alleviate the stress of heavy metals. It serves as a biotechnology and a sustainable and environmentally convenient alternative to enhance agricultural productivity [9].

In recent years, PGPR has garnered significant attention for its role in phytoremediation. According to statistics, the annual number of publications from 1995 to 2023 exhibited an exponential increase (Figure 1a). Notably, the years with high growth rates were 2019 (with a growth rate of 36.33% compared to the previous year) and 2020 (with a growth rate of 31.54% compared to the previous year). Such sharp growth is attributed to the global environmental concern for heavy metal pollution along with the rapid development of industry.

As shown in Figure 1b, the study of interaction of PGPR with various harmful heavy metals (especially Zn and Cd) has also soared in the last 5 years, indicating that considerable efforts have been devoted to the field of heavy metal treatment involving PGPR. PGPR can enhance the absorption and transport of heavy metals in soil plants by supplying nutrients, increasing plant yield, and enhancing stress resistance to heavy metals [10,11]. Utilizing PGPR as a bioinoculant enables nutrient recycling, soil structure stabilization, and the regulation of heavy metal bioavailability and toxicity, thereby promoting plant biomass and root growth [12]. 

PGPR creates a beneficial relationship with plants and has gained widespread interest in the fields of heavy metal stress management in agricultural and environmental research. In this paper, the detoxification mechanism of PGPR to heavy metals in soil is divided into two aspects: promoting plant growth and interacting with heavy metals, which can also be understood as endogenous accumulation and exogenous decreases in heavy metals via plants, respectively. Its recent applications in agricultural and environmental remediation, as well as the potential prospects and limitations of PGPR-assisted phytoremediation, are also discussed.

## 2. Plant-Growth-Promoting Rhizobacteria (PGPR)

The term “Rhizosphere” was first proposed by German microbiologist Lorenz Hiltner in 1904 to explain the impact of plant root exudates on soil microorganisms [13]. Rhizosphere microbes are the community of microbes that inhabit this area, mainly including bacteria, actinomycetes, fungi, algae, and viruses [14,15]. PGPR in the plant rhizosphere typically refers to beneficial, free-living bacteria in the soil that are connected to plant roots [16]. Plants raise the pressure on microorganisms to survive through allelopathy, enabling them to selectively favor bacteria that contribute most to their growth [17]. The concentration of rhizospheric bacteria is particularly high in plants because the roots of most plants can exude significant levels of nutrients (especially tiny molecules like amino acids, sugars, and organic acids), which provide energy for the growth and metabolism of bacteria [18]. 

In 1978, Kloepper and Schroth first introduced the concept of PGPR, defining it as a group of microorganisms that colonize the plant rhizosphere and promote plant growth through the plant rhizosphere [18]. Table 1 shows the typical species and primary mechanisms of PGPR in facilitating heavy metal extraction in plants.

PGPR presented in heavy-metal-contaminated soil can not only promote plant growth and enhance plant biomass, but also interact with heavy metals, thus improving the efficiency of the phytoremediation of heavy-metal-contaminated soil. From these two aspects, we will give a thorough overview of the detoxification mechanisms of PGPR.

## 3. PGPR Can Promote Plant Growth

The primary mechanism underlying its growth-promoting effects is depicted in Figure 2.

### 3.1. Biological Nitrogen Fixation

Nitrogen is not only one of the most important nutrition sources for plant growth, flowering and fruiting, but also an essential element of amino acids, protein, chlorophyll, nucleic acids, membrane lipids, ATP, NADH, co-enzymes, etc. [42,43]. Yet, only active types of oxidized (such as NO_X_) or reduced (such as NH_3_ and amines) nitrogen can be assimilated by plants [44]. The nitrogen needed for crop growth is usually supplemented by the application of nitrogen fertilizer. However, only 30–50% of the nitrogen fertilizer is absorbed by the plants, and the rest enters the aqueous environment, causing water eutrophication [45]. Therefore, finding an environmentally responsible strategy to ameliorate the situation is essential. PGPR acts as a microorganism that can coexist with plants to fix nitrogen; it has great value in agricultural application and environmental restoration. 

Biological nitrogen fixation (BNF) refers to the process of converting nitrogen in the atmosphere into ammonia available to plants through the action of nitrogen-fixing microorganisms [46]. Common nitrogen-fixing microorganisms include *Rhizobium* spp., *Acetobacter* spp., *Arthrobacter* spp., *Citrobacter* spp., *Clostridium* spp., *Streptomyces* spp., and so on [47]. Among them, nitrogen-fixing rhizobia, containing nitrogenase, are a primary type of nitrogen-fixing microorganism, mainly inhabiting the roots of leguminous plants and forming nodules with plant roots. These nodules provide a suitable microenvironment for nitrogenase to convert nitrogen into ammonia to prevent it from being subjected to hypoxia, risking inhibitory activity. Plants provide a good living environment, carbon energy, and essential nutrients for rhizobia, and rhizobia provide nitrogen nutrients for plants [48,49,50]. The symbiosis of rhizobia and legumes is one of the most famous symbiotic relationships in nature. According to research, almost 70% of bio-fixed nitrogen comes from rhizobia and legumes, and rhizobia provide 90% of the nitrogen required by these plants [47]. Inoculation with *Bradyrhizobium* sp. increased both biomass and nitrogen content, confirming this point [10]. PGPR contains the key enzyme nitrogenase of BNF, with its activity typically regulated by the transcription of the nitrogen fixation gene (*nif*) [51]. Nitrogen fixation is instrumental in promoting plant growth, highlighting one of the vital roles of PGPR in enhancing plant vitality. Six common PGPR strains exhibited nitrogenase activity and significantly promoted the growth of wheat and spinach in the study by Cakmakcı et al. [52]. It was also verified that the inoculation of PGPR significantly increased the nitrogenase activity of *Dalbergia sissoo* seedlings and promoted its growth [53].

Typically, heavy metals can adversely affect the growth, nodules, nitrogenase activity, and nitrogen-fixing effects of legumes [54]. This has provided researchers with ideas for the selection of heavy-metal-resistant rhizobia. It was found that excessive amounts of heavy metals such as Cu^2+^ and Zn^2+^ reduced nitrogenase activity and nodule formation in alfalfa, while co-inoculation with alfalfa and rhizobium (*Agrobacterium tumefaciens*) could alleviate heavy metal stress and significantly increase nitrogenase activity and plant biomass [55]. Chen et al. also showed similar results, where inoculation with rhizobia alleviated copper-induced growth inhibition and significantly increased the nitrogen content and copper uptake of the plant [56]. Following their screening of *Burkholderia* spp., a Cd-resistant nitrogen-fixing bacterium, Shen et al. discovered that inoculating plants with *Burkholderia* spp. decreased Cd concentrations in their roots and leaves by 58.11% and 64.54%, respectively, and by 72.89% and 70.03%, respectively, when compared to uninoculated plants [57]. Furthermore, PGPR possesses plant-growth-promoting properties, thereby indirectly enhancing nitrogen bioavailability by promoting root surface area and morphology, resulting in higher nitrogen uptake.

### 3.2. Phosphate Solubilization

Phosphorus stands as a crucial nutrient vital for plant metabolism and nutrient cycling, ranking second only to nitrogen in significance [58,59,60]. Although soil harbors abundant phosphorus in both organic and inorganic forms, it often binds with calcium ions, iron ions, aluminum ions, and others, rendering it challenging for plants to uptake and utilize effectively [59]. To bolster phosphorus nutrition for crops and enhance yields, agricultural chemicals such as phosphorus-based fertilizers are commonly employed. However, excessive phosphorus has caused serious environmental pollution issues, including a decline in soil fertility, an imbalance in the soil microbial system, and water eutrophication [61]. Hence, there is an urgent demand for a cost-effective and environmentally responsible alternative to the use of phosphorus fertilizers.

One of the most promising avenues for developing durable and secure technology is the application of phosphorus-solubilizing bacteria (PSB) [62], which plays an important role in phosphorus cycling and promoting plant growth. It has been demonstrated that PGPR functions as a PSB to transform insoluble phosphorus into a form that plants can use [63], mainly by secreting organic acids (like citric acid, oxalic acid, and gluconic acid), chelating metal ions to form soluble complexes (such as phosphates of calcium, iron, and aluminum), and by producing enzymes (pyridoxal phosphatase (PDXP), phytase, C-P lyase) to hydrolyze organic phosphorus in soil into inorganic forms. The phosphorus-dissolving pathways assisted by PGPR are summarized in Figure 3.

*Pseudomonas* spp., *Enterobacter* spp., *Bacillus* spp., *Serratia* spp., *Arthrobacter* spp., *Burkholderia* spp., *Aspergillus* spp., *Gongronella* spp., *Penicillium* spp., *Talaromyces* spp., and so on, have been identified as PSB [63,64,65]. In addition to providing plants with essential P nutrients, PSB also produce other metabolites, like indole-3-acetic acid (IAA) and siderophore, that are highly effective in promoting plant growth. According to Prakash et al., the PSB *Bacillus* sp. STJP can promote crop growth by generating IAA, siderophores, and phosphorus solubilization [66].

Furthermore, PSB have demonstrated the ability to facilitate the mineralization of heavy metals and generate stable mineralization products [67]. Teng et al. showed that *Pseudomonas* L1–5 could transform Pb^2+^ into lead hydroxyapatite and lead pyroalumite by dissolving phosphorus, showcasing the promising application potential in the bioremediation of lead-contaminated soil [68]. The effectiveness of phosphorus solubilization is influenced by the bacterial species. For instance, when 800 mg/kg of rock phosphate was introduced, *Pantoea* sp. and *Enterobacter* sp. significantly enhanced P solubilization by 49.9% and 88.6%, respectively, which led to a 13.7–26.4% increase in the immobilization of Pb [69].

In conclusion, PSB can relieve heavy metal stress in plants in the following ways:

(i) Secreting organic acids and phosphatases to dissolve insoluble phosphorus and provide necessary phosphorus nutrition for plants to promote plant growth; (ii) secreting IAA and siderophores, which provides the potential to promote plant growth; (iii) inducing heavy metal mineralization to fix heavy metals and reduce their bioavailability.

### 3.3. Siderophore Production

Iron is one of the essential nutrients for plants and is crucial for many cell functions. Despite being widely distributed throughout the Earth’s crust, iron is typically found in an insoluble oxidation state that makes it difficult for plant roots to absorb and use [70]. Under low iron stress, some microorganisms and crops can produce a low-molecular-weight organic molecule called siderophore that has a strong affinity for Fe^3+^ [71]. Therefore, an iron absorption system is formed, with siderophore serving as the primary material: The siderophore chelates with insoluble iron to form a complex, and the membrane receptor transports the complex across the microbial membrane. Following the enzyme system reaction in the cell, the Fe^3+^ bound to the iron carrier is re-released; then, Fe^3+^ is reduced to Fe^2+^, which can be absorbed and utilized by most plants, promoting plant growth [63]. Furthermore, siderophores can also chelate with other toxic metal ions to reduce their concentration. In this regard, siderophores also protect cells from oxidative damage and promote plant growth by preventing oxidative stress. Studies have shown that intracellular reactive species of superoxide (O_2_·) or hydrogen peroxide (H_2_O_2_) can destroy Fe–S clusters, leading to excess free Fe^2+^ released from the damage cluster. The released Fe^2+^ subsequently can be catalyzed by the Fenton reaction to produce harmful hydroxyl radicals (OH·) which can attack DNA, and the siderophores accumulated in cells can chelate the release of Fe^2+^, thus inhibiting the formation of OH· [72,73]. Pyoverdine (PVDI), a fluorescent iron carrier, can protect *Pseudomonas aeruginosa* from oxidative damage by chelating with released ferrous, inhibiting the formation of OH· [73]. A ginseng rhizosphere strain, DCY119T, when ingested by iron-stressed seedlings, can produce siderophores and get rid of surplus ROS, thereby lowering oxidative stress [74]. 

In iron-poor environments, PGPR can outcompete rhizosphere pathogenic microorganisms for scarce iron resources. This competitive advantage inhibits the proliferation and propagation of pathogenic microorganisms, thereby making a substantial contribution to the biological control of fungal infections [70]. Di et al. demonstrated that *Aureobasidium pullulans* L1 and L8 effectively compete with *Monilinia laxa* for iron resources through the secretion of siderophores, diminishing the mycelial growth and conidial germination of *M. laxa*, thus providing protection to peach fruit from post-harvest decay [75]. In addition, compared with plant-derived siderophores, PGPR that can secrete siderophores exhibit greater efficiency in binding Fe^3+^, attributed to their superior solubility and stronger affinity for metals [76].

### 3.4. Phytohormone Production 

As a natural source of phytohormones, PGPR may produce a variety of phytohormones concurrently, which is crucial for promoting root development, nutrient uptake, biomass synthesis, and other characteristics of the host plants [77]. Phytohormones secreted by PGPR, like auxin (IAA), cytokinin (CTK), gibberellin (GA), and ethylene (ETH) [78], make a significant contribution to plant growth under heavy metal stress conditions [79].

#### 3.4.1. Auxin

Auxin is a kind of low-molecular-weight molecule that naturally exists in plants and is involved in processes like organ development, root hair production, and bud elongation [80,81], such as indole-3-acetic acid (IAA) [82,83]. Many PGPR, including *Azospirillum* spp., *Alcaligenes* spp., *Klebsiella* spp., *Enterobacter* spp., *Acetobacter* spp., *Bacillus* spp., *Pseudomonas* spp., *Xanthomonas* spp., *Rhizobium* spp., *Arthrobacter* spp., and *Bradyrhizobium* spp., have been reported to synthesize IAA [84,85]. In heavy metal pollution areas, PGPR can stimulate plant root growth by secreting IAA, which increases the plant root surface area and enhances access to soil nutrients and heavy metal accumulation, so as to improve the efficiency of the plant root absorption of heavy metals [70,86]. Ma et al. found that PGPR increased Cu and Zn content in *Brassica oxyrrhina* by 146% and 61% [86], and the inoculation of the PGPR strain significantly increased the shoot copper content of maize and sunflower [87]. The higher ability of heavy metal accumulation in plants inoculated with PGPR was related to the high level of IAA produced by bacterial strains, which increased the surface area and length of roots, providing plants with more soil nutrients and heavy metals. Thus, bacteria that produce IAA in rhizosphere soil are considered to play an important role in heavy metal uptake by plants. Carlos et al. reported that stress from Pb, Cu, and As significantly increased the amount of IAA synthesized by *Serratia* K120, *Enterobacter* K131, *Enterobacter* N9, and *Escherichia coli* N16, promoting the growth of sunflower plants, which has potential application in phytoremediation systems [88]. An experiment in which researchers injected IAA-deficient mutants into wheat (*Triticum aestivum*) and tracked the growth response showed that wheat infected with IAA-deficient mutants of *P. moraviensis* grew more slowly and presented smaller roots than wild-type strains, verifying the significance of PGPR in encouraging plant development with the production of IAA [89]. 

#### 3.4.2. Cytokinin

Cytokinin (CTK) is a hormone widely found in higher plants, algae, and bacteria [70], and is the second most important phytohormone after IAA [90]. Cytokinin stimulates cell division, cell expansion, and tissue expansion, and plays a vital function in promoting chloroplast development, vascular development, and bud differentiation in plants [70,91,92]. Currently, numerous studies have shown that PGPR can produce cytokinins and promote plant growth under heavy metal stress [93]. For example, Wang et al. inoculated *Arabidopsis thaliana* with a cytokinin-producing PGPR, *Bacillus megalosporum*, and found that elevated transcript levels of the cytokinin receptors in plant root shoots and roots significantly promoted plant growth [94]. According to Piotrowska, cytokinin significantly alleviated the growth inhibition of the green alga *Acutodesmus obliquus* and enhanced its ability to combat lead toxicity under lead stress [95]. Similarly, Yu et al. reported that cytokinin plays a crucial role in increasing the biomass and cadmium uptake efficiency of the super-accumulating plant *Sedum alfredii* [96].

Nieto and Frankenberger demonstrated a significant improvement in the growth state of plants inoculated with *Azotobacter chococcum*. The dry weight of root and shoot tissue was reported to be 5.57 times and 5.01 times higher, respectively, compared to the blank group. They concluded that the increased plant yield was primarily attributed to the generation of cytokinin by rhizosphere bacteria [97]. The application of thiadiuride and kinetin, cytokinin-like agents, on maize (*Zea mays* L.) seedlings can alleviate the deterioration effects of heavy metals on seed germination ability, seedling growth, and membrane permeability [98]. Similarly, the application of kinetin increased the photosynthetic rate of pea plants under Cd stress [99]. Therefore, cytokinins also have an important role in alleviating plant toxicity from heavy metals.

#### 3.4.3. Aminocyclopropane-1-Carboxylate (ACC) Deaminase 

A key regulator of plant growth, ethylene, is a gaseous plant hormone. Low levels of ethylene regulate bud and root growth, encourage flowering, fruit ripening and shedding, and leaf aging and shedding, and significantly influence how plants react to abiotic and biological stimuli [100,101]. Conversely, high levels of ethylene inhibit normal plant growth and promote plant senescence and even premature death [70]. However, plants can experience large amounts of ethylene production under heavy metal stress. Therefore, the proper regulation of ethylene level is crucial for the growth of plants stressed by heavy metal stress [47].

Ethylene is synthesized by two exclusive enzymatic reactions. In the first step, the substrate *S*-adenoyl-_L_-methionine (SAM) is converted by ACC synthetase (ACS) to ACC and 5′-methylthioadenosine (MTA). In the second step, ACC is converted to ethylene, CO_2_, and cyanide by ACC oxidase (ACO). Among them, ACC is a direct precursor of ethylene, so ethylene levels can be controlled by regulating ACC levels [102]. If reducing the activity of ACS and ACO, ethylene synthesis could be inhibited to some extent. In the Misra study, inoculation with PGPR significantly reduced the ACS and ACO activity of *Zea mays* [103]. The inoculation of wheat plants with the PGPR strain *Anospira brasiliensis* FP2 resulted in a drop in ACO expression in vivo, resulting in the reduced ethylene content of wheat roots [104]. Similarly, volatile organic compounds emitted by *Bacillus subtilis* SYST2 repressed the transcription of the *ACO1* gene and caused a decrease in endogenous ethylene content in tomato seedlings [105]. 

Moreover, there is an enzyme that can decompose ethylene, making a great contribution to reducing ethylene content, namely ACC deaminase (ACCD). This enyzme can decompose the ethylene precursor ACC into α-ketobutyrate and ammonia [47,106,107]. Glick et al. concluded from model prediction analysis that almost all PGPR have ACC deaminase activity [108]. Therefore, PGPR inoculation has a positive impact on the accumulation of heavy metals in plants. Sunflowers grew better and accumulated more Zn and Pb in the presence of PGPR (*Bacillus safensis* FO-036b(T) and *Pseudomonas fluorescens* p.f.169), which contains ACC deaminase, according to research by Mousavi et al. [109]. Carlos et al. discovered that the rhizobium strains of *Enterobacter* N9, *Serratia* K120, *Klebsiella* Mc173, and *Escherichia coli* N16 that produce ACCD dramatically lengthened cauliflower’s shoots and roots, thus accelerating the phytoremediation of metals such as copper, nickel, zinc, lead, and arsenic [88]. In summary, it is evident that PGPR, particularly those capable of producing ACC deaminase, play a significant role in aiding plant growth and improve the accumulation of heavy metals.

#### 3.4.4. Gibberellins

Bioactive gibberellin (GA) is a diterpenoid plant hormone that is involved in several plant developmental processes, including seed dormancy, germination, flowering, fruit ripening, root growth, and root hair enrichment, through complex biosynthesis [70,110,111]. At present, more than 130 kinds of GAs have been found. PGPR mainly produce GA1, GA3, GA4, and GA20, with GA3 being the most prevalent form [70,112]. Typical PGPR species that can produce GA include *Acetobacter* spp., *Bacillus* spp., and *Azotobacter* spp. [70,113].

GA can improve plant adaptation to heavy metal toxicity. One study showed that the application of GA3 alleviated the toxicity of Cu stress to pea (*Pisum sativum* L.) seedlings [114]. Under Cd stress, GA application on *Cyphomandra betacea* promoted beetroot seedling growth and increased the biomass, leaf net photosynthetic rate, and carotenoid and soluble sugar content. The Cd content of *C. betacea* seedlings reduced gradually with increasing concentrations of GA [115]. For hyperaccumulator *Sedum alfredii,* the application of GA significantly increased the dry biomass of the root, stem, leaf, and shoot. The enhanced accumulation of Cd and Pb in the shoot of *S. alfredii* demonstrates significant potential for heavy metal phytoremediation [116].

## 4. Interactions between PGPR and Heavy Metals 

### 4.1. Chelation

#### 4.1.1. Exopolysaccharide Production

Exopolysaccharides (EPSs) are a kind of high-molecular-weight natural polymer secreted by microorganisms, and include sugars, proteins, amino acids, lipids, and other substances. They typically build up on the surface of bacterial cells and serve as a barrier against harmful external factors such as phagocytosis, pathogen attachment, drying, pH, and heavy metals [78,117,118]. According to numerous studies, EPSs can reduce toxicity by adsorbing heavy metals through electrostatic contact, ion exchange, complexation, surface precipitation, redox, and other interactions [119,120]. The mechanism illustrating how EPSs protect cells from heavy metal ions is depicted in Figure 4. Without EPS protection, metal ions will directly contact the cell and react with specific proteins or enzymes within the cell, causing a loss of their activity and eventual cell damage. 

Electrostatic interactions between functional groups like hydroxyl, acetylamino, and amino groups in EPSs and positively charged metal cations (Cd^2+^, Pb^2+^, Ni^2+^, Co^2+^, and Cr^5+^) result in the formation of organometallic complexes on the surface of cells [120]. Ion exchange is a popular mechanism to explain the metal bio-sorption by EPSs [121,122]. The heavy metals chromium and cadmium form a combination with the EPSs generated by PGPR (*Azotobacter* spp.), which decreases their ability to migrate and lessens the pressure on wheat growth. Higher pH values typically result in fewer protons in solution being used to compete with metals for binding locations in EPSs, leading to increased metal adsorbing in the ion exchange equilibrium [123]. Additionally, EPSs can play a role in mitigating the toxicity of heavy metals through surface precipitation or redox reactions. The chemical properties of the solution, especially the pH level, have a profound impact on the shape and solubility of metals in aqueous solutions. As the pH rises, most metal ions undergo transformation from hydrated metal cations to hydroxylated monomers and polymers. Over time, this process leads to the formation of crystalline oxide precipitation [124,125].

It has been reported that EPS components facilitate the enzymatic reduction in Cr (VI) by interacting with the toxic group of carboxyl and hydroxyl groups, thereby binding Cr (VI). Additionally, protein components within EPSs may also contribute to reductions in Cr (VI). Furthermore, primary functional groups, such as COO- and -OH groups in EPSs, can bind Cr (VI) to shield cells from the toxic effects of Cr (VI) [126,127]. EPSs may also resist toxic metals by trapping them outside cells or forming biofilms. These two pathways can effectively mitigate the environmental toxicity of metals and enhance the symbiotic development and growth of legumes in metal-contaminated soils [128].

#### 4.1.2. Metallothionein Production 

Metallothioneins (MTs) are a family of universal low-molecular-weight, cysteine-rich (about a third of their amino acid content) proteins [129], widely distributed in eukaryotes such as fungi, plants, animals, and some prokaryotes [130,131]. They can form sulfur-based metal clusters by binding to a large number of metal ions, such as Zn^2+^, Cd^2+^, and Cu^2+^, through the thiol group of their cysteine residues, which play an important role in regulating the dynamic balance and detoxification of plant metal ions [132,133].

Recombinant strains with MTs were found to contribute to plants binding heavy metals in soil and acted as free-radical scavengers [134]. Murthy et al. treated plants with PGPR strains with bacterial MTs and found that plants inoculated with PGPR enhanced the bioremediation process in metal-infected soils and had positive results for the removal of heavy metals such as Pb^2+^ [135]. In tobacco leaves, MTs can regulate the responses of Silene to Cu stress [136].

#### 4.1.3. Soil Organic Acid Production

Some PGPR can secrete organic acids, including formic acid, acetic acid, tartaric acid (TA), succinic acid, oxalic acid (OA), citric acid (CA), and gluconic acid (GA) [137,138]. For some super-enriched plants, the release of organic acids is one of the mechanisms for the migration of heavy metals (by altering their mobility) [139]. For instance, soil PSB strains can secrete large amounts of gluconic acid, thereby improving cadmium mobility and increasing the bioavailability [140]. Organic acids can also enhance the metal mobility in soil by reducing soil pH and forming complexes with heavy metals [141], thus increasing the absorption of heavy metals in the rhizosphere soil by some plants. Previous studies have shown that low-molecular-weight organic acids (LMWOAs) secreted by plant roots can form soluble Cd-LMWOA complexes with Cd, making it more easily absorbed by plants [142]. Similarly, treatment with tartaric acid, malic acid, oxalic acid, and citrate acid considerably raised root and stem Cd concentrations compared to the control, having a favorable impact on the uptake of total Cd in the soil [143]. Some studies used *Brassica juncea* L. to explore the effects of different doses of CA on heavy metal accumulation and stress tolerance and showed that the exogenous application of CA in growth medium containing Cd significantly reduced the harmful effects of Cd on plants [144].

As early as 1998, Hassen et al. reported that the presence of citric acid increased the biosorption of Cr by *Pseudomonas aeruginosa* and Cu by *Bacillus thuringiensis* [145]. Farid et al. found that the combined application of citric acid and 5-aminolevulinic acid could improve the biomass, photosynthesis, and gas exchange characteristics of sunflowers in Cr-contaminated soil [146]. Chai et al. showed that the content of oxalic acid in the inflorescence, stem, and fine root of *Saussurea involucrata* was positively correlated with the bioaccumulation of Cd [147]. Similarly, Chen et al. found that tartaric acid or malic acid can effectively improve the growth potential of hybrid flowers under Cd stress [148]. Tomato seedlings inoculated with *Pseudomonas aeruginosa* and *Burnetidia eriagladioli* reduced Cd-induced toxicity by upregulating the levels of LMWOAs (fumaric acid, malic acid, succinic acid, and citric acid), which further proves the important role of PGPR in the secretion of organic acids in plants to alleviate heavy metal stress [149].

### 4.2. Induced Systematic Resistance

Induced systematic resistance (ISR) is a state of enhanced defense formed in plants by activating potential resistance, and is induced by various factors, such as rhizosphere bacteria [150]. ISR initiates multiple potential defense mechanisms, including increasing the activity of chitinase, β-1,3 glucanase, and peroxidase; accumulating antimicrobial low-molecular substances such as phytolexins; and forming protective biopolymers, viz. lignin, callose, and hydroxyproline-rich glycoprotein [151]. Given that the efficacy of ISR hinges on defense mechanisms triggered by inducers, employing natural PGPR strains as inducers of plant defense responses could enhance their practicality and effectiveness.

It was found that PGPR can complete the ISR process by increasing the activity of antioxidant enzymes [152,153]. As shown in Figure 5, under heavy metal stress, plant cells will release excessive ROS, resulting in oxidative damage. However, PGPR promote the synthesis of a variety of antioxidant enzymes by plant cells to resist this oxidative damage, such as superoxide dismutase (SOD), peroxidase (POD), catalase (CAT), ascorbic peroxidase (APX), etc. Thich can effectively remove ROS from plant cells and is conducive to plant growth [154]. Gururani et al. isolated two PGPR strains from the potato rhizosphere, *Bacillus pumilus* strain DH-11 and *Bacillus firmus* strain 40, which improved the Zn tolerance of potato by up-regulating the gene transcription levels of ROS-scavenging enzymes (APX, SOD, CAT, DHAR, and GR) in potato plants, contributing to increased plant tolerance to Zn [155]. According to Ju et al., PGPR and rhizobium-inoculated alfalfa could decrease the accumulation of ROS, increase the activity of antioxidant enzymes, and enhance the copper tolerance of alfalfa [156]. Sunflower and tomato can increase the synthesis of SOD and CAT, reduce the production of ROS, and enhance Cr tolerance after being inoculated with *Pseudomonas* sp. [157]. PGPR significantly improved the antioxidant system produced by Cd induction by upregulating the mRNA expression of SOD, POD, and PPO genes in tomato [158].

### 4.3. Transform Toxic Heavy Metals 

Bacteria in the soil can convert toxic heavy metals into forms that are easily absorbed by plant roots [159], thereby improving the efficiency of phytoremediation. PGPR can enhance the bioavailability of heavy metals, transforming them from insoluble forms to soluble forms [160]. This process is also known biodegradation [161]. For example, the conversion of selenate to organic Se in the presence of bacteria can increase selenium accumulation in plants [162]. Jeong et al. showed that *Bacillus megaliium* significantly increased the amount of exchangeable Cd in rhizosphere soil [163]. The inoculation of sunflower with *Micrococcus* sp. MU1 and *Klebsiella* sp. BAM1 (Cd-tolerant PGPR strain) could increase the concentration of water-soluble Cd in rhizosphere soil, thus enhancing the absorption of Cd by sunflower roots [164]. For instance, biosurfactants produced by *Pseudomonas aeruginosa* BS2 lead to the solubilization of Pb and Cd [165]. *Bacillus subtilis*, *B. cereus*, *Flavobacterium* sp., and *Pseudomonas aeruginosa* increased the availability of water-soluble Zn in soil and Zn accumulation by plants [166].

## 5. Conclusions and Future Perspectives

Heavy metals have led to severe environmental pollution problems, prompting extensive research into heavy metal detoxification methods. Considering the economic and environmental benefits, the PGPR-assisted phytoremediation of heavy metal pollution emerges as a promising environmentally friendly strategy. By fixing nitrogen, dissolving phosphorus, secreting plant growth hormones, and increasing the bioavailability of heavy metals, PGPR can stimulate plant growth through their metabolic processes, enhancing the effectiveness of soil heavy metal remediation. At the same time, PGPR affect plant cell physiological activities, induce ISR, activate plant antioxidant enzymes, increase iron supply by secreting high-affinity siderophores, competitively inhibit the uptake of heavy metals by plant roots, and influence the absorption, transportation, and intracellular distribution of heavy metals. These actions help alleviate heavy metal stress, enhance heavy metal tolerance, and ultimately improve the efficiency of phytoremediation. Under various heavy metal stress environments, PGPR play an important role in the migration and transformation of heavy metals in rhizosphere soil and in helping plants to absorb and accumulate heavy metals or immobilize the soil. 

Future research directions for PGPR include the following: further expanding the PGPR strain bank; devising strategies for the synergistic use of multiple PGPR strains; sourcing mixtures of multiple PGPR strains directly from natural environments; and developing macromolecular materials such as biochar combined with PGPR for phytoremediation. Moreover, there is potential to broaden the application scope of PGPR to areas such as sludge treatment, sewage purification, mine reclamation, sediment restoration, and more. It is imperative for researchers to continue advancing their understanding of PGPR and promptly translate this knowledge into practical applications.

## 6. Problems in the Practical Application of PGPR in Environmental Remediation

In view of the previous research progress and looking forward to research work in the future, more attention should be paid to the following aspects.

First, the use of PGPR in agricultural management is constrained by factors like short survival times, low survival rates, and the uniqueness of effective strains in actual environments. Mathematical modeling and computer-based simulation results indicate that the competition for limited resources between PGPR populations and resident microorganisms is the most important factor for determining the survival of PGPR. The most effective PGPR application is in organic and mineral-poor soil or stressed soil, because the development of resident microflora is inhibited [167]. Currently, the majority of PGPR utilized in research are isolated from contaminated areas, which is time-consuming and laborious if they need to be separated and screened from the target area before each governance. This greatly limits the application of PGPR. It is worth noting whether high-performance strains can be developed in the future, or introducing other organisms or vectors to enhance the survival rate and efficiency of PGPR.

Second, given that various strains of plants have varying capacities for tolerating certain heavy metals, care should be taken while applying medications in real agricultural applications to prevent unneeded losses. To maximize their tolerance to heavy metals through domestication, multiple PGPR repair combinations should be developed to receive multiple PGPR mixtures directly from the environment.

Third, current studies primarily focus on the individual management of heavy metal stress within laboratory settings. However, actual heavy-metal-polluted land is often accompanied by chemical pollution, drought, salt stress, and other harsh conditions. Moreover, plant hormone regulation and interactions vary across different stress environments. Therefore, future research on the mechanism of PGPR in improving plant resistance to heavy metal toxicity should be further explored in field studies, and the different responses of plants under common stresses of two or more kinds of pollution should be focused on. And on the basis of screening excellent functional PGPR strains, the development of special PGPR strains suitable for different crops, different geologies, and different temperatures could play an important role in dealing with different soil types and climatic conditions, which is the direction and goal of PGPR strain research and development.

Fourth, there is currently limited research on the long-term outcomes for plants after the phytoremediation of heavy metal pollution. Most studies tend to focus on the restored soil, which represents a drawback of phytoremediation. Certain heavy metals like As, Se, and Hg can be absorbed by accumulator plants and subsequently volatilized into the atmosphere as gases, a process known as phytovolatilization [168]. In this process, the metal content in plants remains relatively low, suggesting a long-term remediation strategy. However, for plants unable to undergo phytovolatilization to accumulate excessive heavy metals, incineration might be the optimal solution, by which heavy metal oxides can be recovered and plant biochar—an environmentally friendly material—can be produced, thus achieving a green cycle.

Other suitable application of PGPR should also be paid attention to. PGPR are employed to safeguard plants from pathogens and mitigate environmental stress through a range of biological control mechanisms, including the secretion of antibiotics, the induction of systemic resistance, and the induction of systemic tolerance, among others. In agricultural production, plant diseases exert a serious impact on the growth and development of plants. However, the adoption of this technology is currently limited, primarily due to a lack of awareness among most farmers and variations in farmland practices across different locations [169]. In light of the aforementioned discrepancies, researchers need to devise more user-friendly strategies for the application of PGPR. Farmers are more prone to employ the process if it is straightforward and demonstrably effective.

In conclusion, PGPR are anticipated to replace chemical fertilizers, insecticides and synthetic growth regulators in the future, thereby fostering sustainable agricultural growth and facilitating the bioremediation of heavy metal contamination in the environment. 

## Figures and Tables

**Figure 1 microorganisms-12-00700-f001:**
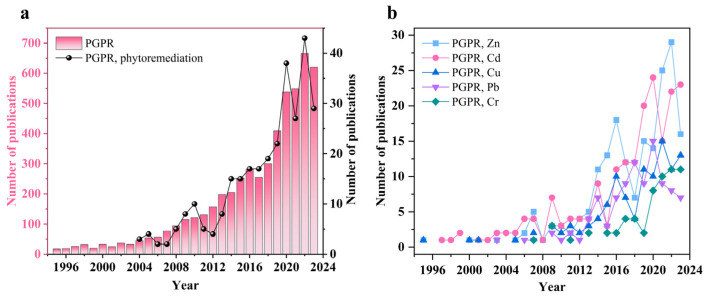
Number of papers on PGPR included in Web of Science from 1995 to 2023. (**a**) Search for the keywords “PGPR” and “PGPR, phytoremediation”. (**b**) Search for keywords “PGPR, Zn”, “PGPR, Cd”, “PGPR, Cu”, “PGPR, Pb”, and “PGPR, Cr”.

**Figure 2 microorganisms-12-00700-f002:**
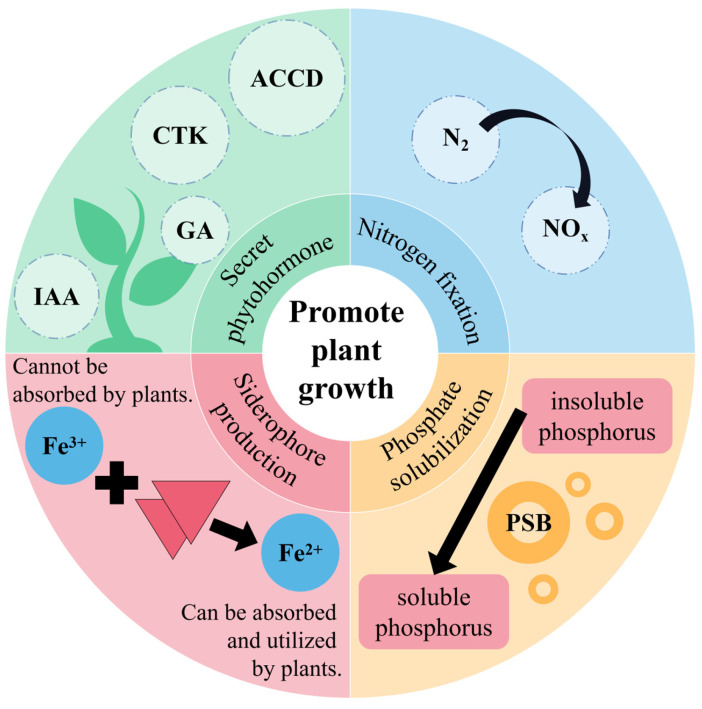
Growth-promoting mechanisms of PGPR.

**Figure 3 microorganisms-12-00700-f003:**
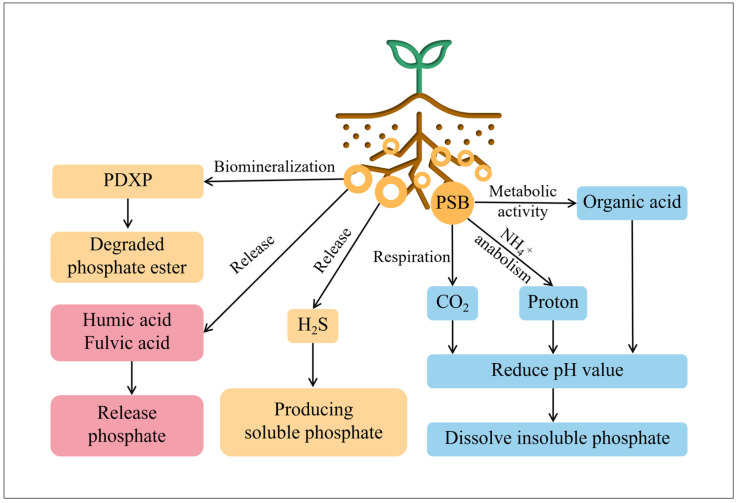
Mechanism of PSB in bio-dissolving insoluble phosphates.

**Figure 4 microorganisms-12-00700-f004:**
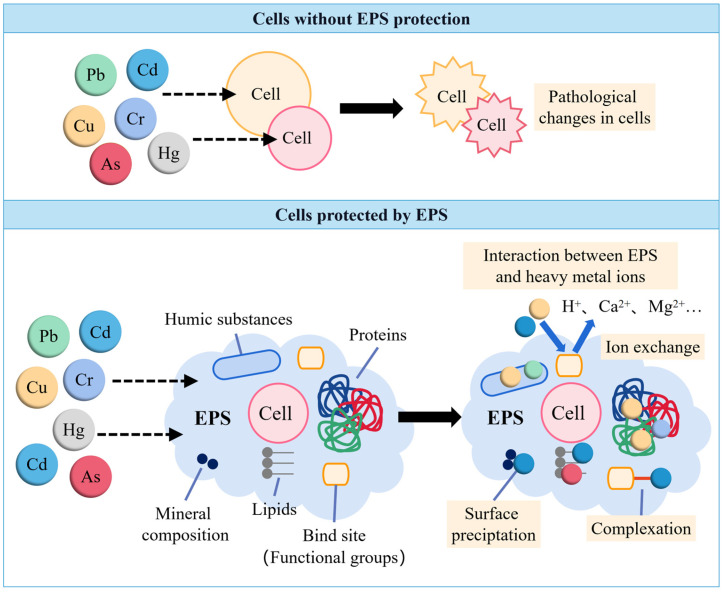
Schematic diagram of mechanism of metal–EPS interactions.

**Figure 5 microorganisms-12-00700-f005:**
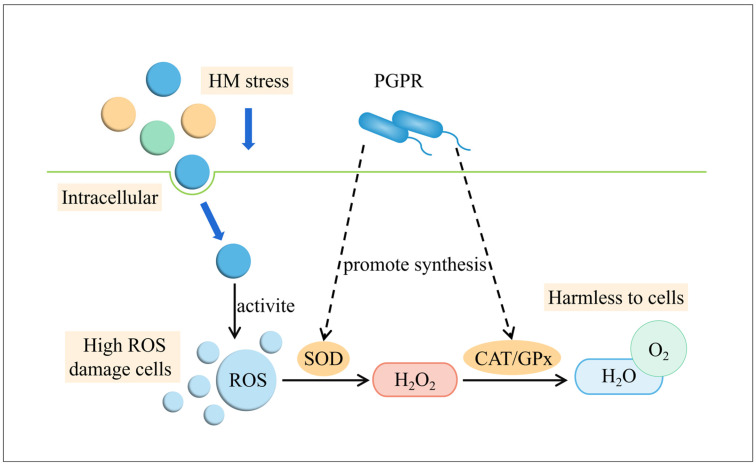
Systemic resistance induced by PGPR.

**Table 1 microorganisms-12-00700-t001:** How PGPR assists in the phytoextraction of heavy metals.

Target Heavy Metals	PGPR	Test Plants	Plant-Growth-Promoting Traits/Mechanisms	References
Cd, Zn	*Bacillus* sp.	*Oryza sativa*	Secretes indole-3-acetic acid (IAA), 1-aminocyclopropane-1-carboxylate (ACC) deaminase, and siderophores; phosphate solubilization.	[19]
Ni, Pb, Cd and Cr	*Pseudomonas putida*	*Zea mays*	Increases the availability of Fe, Zn, K, and Ca.	[20]
Cd	*Bacillus megaterium*	Hybrid *Pennisetum*	Secretes IAA, siderophores, ACC deaminase (ACCD); phosphate solubilization and nitrogen fixation.	[21,22]
Cr	*Pseudomonas* sp.	*Medicago sativa*	Secretes IAA and siderophores; produces ammonia, cellulase, pectinase, chitinase, and ACCD; phosphate solubilization; nitrogen fixation.	[23]
Cd, Pb	*Enterobacter bugandensis* and *Bacillus megaterium*	*Lactuca sativa* L.	Secrete IAA and siderophores.	[24]
Cr	*Agrobacterium fabrum*	*Zea mays*	Secretes siderophores, IAA, and potassium; phosphate solubilization.	[25]
Ca	*Bacillus* spp.	*Brassica napus* L.	Secretes IAA, siderophores and ACCD; phosphate solubilization.	[26]
Pb	*Luteibacter* sp. and *Variovorax* sp.	*Lathyrus sativus* L.	Secrete IAA, siderophores, and HCN; phosphate solubilization.	[27]
Cr	*Klebsiella* sp.	-	Secretes IAA, ammonia, siderophores, and HCN.	[28]
Cd, Pb and As	*Klebsiella michiganensis*	*Oryza sativa*	Secretes IAA and ACCD; nitrogen fixation; phosphate solubilization.	[29]
As, Cd and Cr	*Pseudomonas* sp.	*Lens culinaris*	Secretes IAA.	[30]
Cu	*Kocuria* sp.	*Saccharum spontaneum*	Secretes IAA, product ammonia, and hydrogen cyanide (HCN); phosphate solubilization.	[31]
Ni	*Bacillus* spp.	*Althea rosea* L.	Secretes IAA; siderophore production; phosphate solubilization.	[32]
Cr, Cd	*Azotobacter* sp.	*Lepidium sativum*	Solubilizing of phosphorus; improves the dissolution and retention of iron in the growth medium; nitrogen fixation; produces plant hormones.	[33]
Cd	*Azotobacter* sp.	*Triticum aestivum* L.	Secretes IAA and ACCD; nitrogen fixation; phosphate solubilization.	[34]
As	*Exiguobacterium* sp.	*Vigna radiata*	Secretes IAA and EPS.	[35]
Ni, Zn and Fe	*Psychrobacter* sp. and *Pseudomonas* sp.	*Brassica juncea* and *Ricinus communis*	Secrete siderophores, ACCD, and IAA; phosphate solubilization.	[36]
Cd	*Bradyrhizobium* sp.	*Lolium multiflorum* Lam.	Secretes IAA, siderophores, and ACCD; phosphate solubilization.	[37]
Zn	*Pseudomonas aeruginosa*	*Triticum aestivum* L.	Secretes IAA, ACCD, and siderophores; phosphate solubilization.	[38]
Zn	*Proteus mirabilis*	*Zea mays*	Secretes IAA, siderophore, and ACCD; phosphate solubilization.	[39]
Cd	*Ochrobactrum* sp.	*Oryza sativa*	Secrete siderophores and ACCD.	[40]
Pb, As	*Bacillus* sp.			
Ni	*Bacillus subtilis*	*Brassica juncea*	Secretes IAA; phosphate solubilization.	[41]

## Data Availability

Not applicable.

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
