# Peer review of "Role of Plant-Growth-Promoting Rhizobacteria in Plant Machinery for Soil Heavy Metal Detoxification"

_microorganisms, 2024, doi:10.3390/microorganisms12040700_

Round 1
Reviewer 1 Report
Comments and Suggestions for Authors
The article revises the different mechanisms of PGPR in metal detoxification.
There are a few main concerns about the article:
The language needs to be revised deeply, improve the scientific writing.
There are dome misunderstandings when you speak about phytoremediation. It not always is phytoextraction, and PGPR also can help to immobilize metals in soil, helping the phyostabilsation. Revise all these through the ms, and clarify each example for phytoextraction or phytostabilsation. The first one in the abstract.
Summary briefly each case study you present, main results. Avoid to give too many details. Reference is enough.
Line 33. Remove seriously degraded. What does it mean?
Line 35. Remove so on
Line 47. “Eliminate organic toxic pollutants”. You cannot metlas
Line 51. Remove repair, use another term as Contribute or something similar
Line 77. You say “Graphical abstract”. I have not seen it.
Line 95. Wha was Kloepper definition? Summarize paragraph line 95-102.
Lone 107-110. PGPR can also as you say after, chelate metals reducing the mobility. You have to add this part also, and consider always phytoextraction and phyostabilsation
Lines 117. Star with the main function of N : the most important nutrition….” And them the rest
Summarize lines 121-126
Line 128: “The process of converting N2 from air to ammonia”. Change this way
Lines 130-135. Summarize
Line 136-141. Rewrite. It is no clear the process
Line 158-163. Main data of each article
Remove last sentence (lines 197-199)
Line 200-2006. Rewrite. All paragraph quite confusing
Line 222-224. Rewrite
Line 230. Start the sentence with “in this regard”
Line 230-278. Organize ideas and summarize
Line 250. Remove this sentence, not necessary
Line 268. “rape of inoculation????”
Line 269. In which part of the plant accumulate Cu? This is important
Lines 274-279, change by “an experiment, in which they injected IAA-deficient mutants into wheat (Triticum aestivum) and tracked the growth response showed that wheat infected with IAA-deficient mutants of P. moraviensis grew more slowly and presented smaller roots than wild type strains. [89].
Line 185-288. Rewrite and shorten this lines. Same lines 289-297
Same for section 3.4.3
Line 396-398. Rewrite
Lines 416-428. Summarize; Same lines 442-453, same lines 462-475
Lines 457. What doy you mean? Rewrite
Line 459-460. Check sentence
line 477. Remove it
Lines 489-492. Change by: Under various heavy metal stress environments, PGPR plays an important role in the migration and transformation of heavy metals in rhizosphere soil and in helping plants to absorb and accumulate heavy metals or immobilize in the soil”. Don’t forget this aspect
Line 502-506. rewrite
Comments on the Quality of English Languageit needs to be revised
Author Response
Response to Reviewer 1 Comments
Comments and Suggestions for Authors
The article revises the different mechanisms of PGPR in metal detoxification.
There are a few main concerns about the article:
Point 1: The language needs to be revised deeply, improve the scientific writing.
Response 1: We sincerely thank the reviewer for the thoughtful review and constructive feedback. The manuscript has been thoroughly revised on both language and readability. We really hope that the language level has been substantially improved and can meet the journal’s standard.
Point 2: There are dome misunderstandings when you speak about phytoremediation. It not always is phytoextraction, and PGPR also can help to immobilize metals in soil, helping the phyostabilsation. Revise all these through the ms, and clarify each example for phytoextraction or phytostabilsation. The first one in the abstract.
Response 2: Yes, we completely concur with the reviewer's viewpoint that phytoremediation entails both phytostabilization and phytoextraction, and that PGPR is crucial to both of these processes, as shown in the graphical abstract (Page 1).
In this manuscript, from the perspective of PGPR-assisted phytoremediation, we discussed the significant role of PGPR in the processes of both phytoextraction and phytostabilization in phytoremediation: 1) by stimulating plant growth and aiding plants in the acclimation of environmental heavy metal, PGPR can enhance the efficiency of heavy metal uptake by plants; 2) by interacting with heavy metals in the rhizo-sphere, PGPR can assist in plant stabilization and alleviate heavy metal stress.
In order to highlight the role of PGPR in the detoxification of heavy metals, the structure of the article was not divided by phytoextraction and phyostabilzation, but by the endogenous accumulation and exogenous reduction of heavy metals. We divided the detoxification mechanism into two parts: PGPR-mediated promotion of plant growth and interaction with heavy metals (Graphical abstract, Page 1). To prevent ambiguity and conceptual confusion, we have tried to minimize the occurrence of phytostabilization and phytoextraction in the text. And we have also modified the text accordingly in lines 79-84 (Page 2).
Point 3: Summary briefly each case study you present, main results. Avoid to give too many details. Reference is enough.
Response 3: We thank the reviewer for the comment. We have checked the manuscript throughout and simplified the case descriptions appropriately in the revised manuscript. (Lines 144-145, Page 6; Lines 224-226, Page 8; Lines 454-456, Page 13; Lines 465-466, Page 13)
Point 4: Line 33. Remove seriously degraded. What does it mean?
Response 4: We apologize for the misunderstanding. In the revised manuscript, “seriously degraded” has been removed as suggested, and changed into “The soil polluted by heavy metals shows ecological imbalance...” (Lines 35-36, Page 2)
Point 5: Line 35. Remove so on
Response 5: We thank the reviewer for the comment. The words "and so on" has been deleted as suggested. (Line 38, Page 2)
Point 6: Line 47. “Eliminate organic toxic pollutants”. You cannot metlas
Response 6: We apologize for the mistake. Heavy metal pollution belongs to inorganic pollution, so we removed the word "organic" from the manuscript (Line 50, Page 2).
Point 7: Line 51. Remove repair, use another term as Contribute or something similar
Response 7: We thank the reviewer for the suggestion. In the revised manuscript, we have replaced “plant repair” with “phytoremediation”. (Line 53, Page 2)
Point 8: Line 77. You say “Graphical abstract”. I have not seen it.
Response 8: We have added the Graphical abstract in the revised manuscript. (Page 1)
Graphical abstract
Point 9: Line 95. Wha was Kloepper definition? Summarize paragraph line 95-102.
Response 9: We thank the reviewer for the comment. To make it clearer, the introduction of PGPR has been modified as follows: “In 1978, Kloepper and Schroth first introduced the concept of PGPR, defining it as a group of microorganisms that colonize the plant rhizosphere and promote plant growth through the plant rhizosphere.” (Lines 97-99, Page 3)
We also simplified this paragraph as suggested. (Lines 97-100, Page 3)
Point 10: Line 107-110. PGPR can also as you say after, chelate metals reducing the mobility. You have to add this part also, and consider always phytoextraction and phyostabilsation
Response 10: We thank the reviewer for this suggestion. Yes, PGPR-assisted phytoremediation can be accomplished in two ways: promoting plant growth and interacting with heavy metal. Hence, we have modified the text to sum up these two detoxification mechanisms. (Lines 79-84, Page 3)
As mentioned in Response 2, in order to highlight the role of PGPR in the detoxification of heavy metals, the structure of the article was not divided by phytoextraction and phyostabilzation, but by the endogenous accumulation and exogenous reduction of heavy metals. We divided the detoxification mechanism into two parts: PGPR-mediated promotion of plant growth and interaction with heavy metals (Graphical abstract, Page 1). To prevent ambiguity and conceptual confusion, we have tried to minimize the occurrence of phytostabilization and phytoextraction in the text. And we have also modified the text accordingly in lines 79-84 (Page 2).
Point 11: Lines 117. Start with the main function of N: the most important nutrition….” And them the rest
Response 11: Following the suggestion, we have modified this sentence and started with the main function of nitrogen: “Nitrogen is not only one of the most important nutrition sources for plant growth, flowering and fruiting, ...”. (Lines 114-115, Page 5)
Point 12: Summarize lines 121-126
Response 12: We thank the reviewer for the comment. We have summarized this paragraph and rewrote it, merging it with the first paragraph of section 3.1 together. (Lines 118-124, Page 5)
Point 13: Line 128: “The process of converting N2 air to ammonia”. Change this way
Response 13: We thank the reviewer for the careful checks. This sentence has been corrected as recommended: “Biological nitrogen fixation (BNF) refers to the process of converting nitrogen in the atmosphere into ammonia available to plants through the action of nitrogen-fixing microorganisms.” (Lines 125-127, Page 5)
Point 14: Lines 130-135. Summarize
Response 14: We have summarized and streamlined this paragraph as suggested, incorporating the point 15 and the suggestions from reviewer 2, we have added some new discussions regarding nitrogenase and fixation genes. (Lines 125-145, Pages 5-6)
Point 15: Line 136-141. Rewrite. It is no clear the process
Response 15: We apologize for this unclear description. In the revised manuscript, we have rewritten this section and merged it with the preceding paragraph. (Lines125-145, Pages 5-6)
Point 16: Line 158-163. Main data of each article
Response 16: We thank the reviewer for the comment. In the revised manuscript, the text has been modified as: “Following their screening of Burkholderia spp., a Cd-resistant nitrogen-fixing bacterium, Shen et al. discovered that inoculating plants with Burkholderia spp. decreased Cd concentrations in their roots and leaves by 58.11% and 64.54%, respectively, and by 72.89% and 70.03%, respectively, when compared to the uninoculated plants.” (Lines 154-158, Page 6)
In order to avoid literature redundancy, we have removed unnecessary literature cases.
Point 17: Remove last sentence (lines 197-199)
Response 17: We have removed it as suggested.
Point 18: Line 200-206. Rewrite. All paragraph quite confusing
Line 222-224. Rewrite
Response 18: We apologize for the unclear description. These sentences have been completely rewritten to hopefully make it easy to comprehensible. (Lines 191-199, Page 7; Lines 217-238, Page 8)
Point 19: Line 230. Start the sentence with “in this regard”
Response 19: Thanks for the suggestions of the reviewer, we modified the sentence as suggested, as: “In this regard, siderophores also protect cells from oxidative damage and promote plant growth by preventing oxidative stress.” (Line 218, Page 8)
Point 20: Line 230-278. Organize ideas and summarize
Response 20: We have summarized and reorganized this section to make it clearer and read more smoothly. (Lines 257-291, Pages 8-9)
Point 21: Line 250. Remove this sentence, not necessary
Response 21: We have removed the sentence as suggested.
Point 22: Line 268. “rape of inoculation????”
Response 22: We thank the reviewer for pointing out this mistake. In the revised manuscript, we have changed the word “rape” to “Brassica oxyrrhina” and checked the full text thoroughly. (Line 258, Page 8)
Point 23: Line 269. In which part of the plant accumulate Cu? This is important
Response 23: We agree with the reviewer that the part of the plant accumulating Cu is important. Unfortunately, it was not specifically discussed in this reference. In order to make this point clear, we have replaced it with another literature ([88]Current microbiology, 2021, 78, 1335-1343) which clearly stated the accumulation sites of heavy metal. The text was modified as: “Ma et al. found that PGPR increased Cu and Zn content in Brassica oxyrrhina by 146% and 61% and the inoculation of the PGPR strain significantly increased the shoot copper content of maize and sunflower.” (Lines 258-260, Page 8)
Point 24: Lines 274-279, change by “an experiment, in which they injected IAA-deficient mutants into wheat (Triticum aestivum) and tracked the growth response showed that wheat infected with IAA-deficient mutants of P. moraviensis grew more slowly and presented smaller roots than wild type strains. [89].
Response 24: We have changed the text as suggested. (Lines 267-271, Page 9)
Point 25: Line 185-288. Rewrite and shorten this lines. Same lines 289-297. Same for section 3.4.3
Response 25: We have rewritten the sections mentioned and simplified them appropriately.
(Lines 273-285, Page 9; Lines 286-290, Page 9; Lines 310-317, Page 10; Lines 318-322, Page 10)
Point 26: Line 396-398. Rewrite
Response 26: Thanks to the reviewer for the suggestion. We carefully examined and revised this paragraph and added a new literature to support it ([135]Journal of hematology & oncology, 2018, 11, 1-20). (Lines 392-395, Page 12)
Point 27: Lines 416-428. Summarize; Same lines 442-453, same lines 462-475
Response 27: We have rewritten the sections mentioned above and simplified them appropriately.
(Lines 390-395, Page 12; Lines 406-412, Page 12; Lines 426-430, Page 12; Lines 432-439, Pages 12-13; Lines 460-472, Page 13)
Point 28: Lines 457. What do you mean? Rewrite
Response 28: We sincerely thank the reviewer for careful reading. As suggested by the reviewer, we have checked and modified the sentence to make it clear. (Lines 445-446, Page 13)
Point 29: Line 459-460. Check sentence
Response 29: We have checked the sentence and rewritten it. (Line 449, Page 13)
Point 30: line 477. Remove it
Response 30: We simplified the sentences and removed the unnecessary text: “Jeong et al showed that B. megaliium significantly increased the amount of exchangeable Cd in rhizosphere soil.”(Lines 465-466, Page 13)
Point 31: Lines 489-492. Change by: Under various heavy metal stress environments, PGPR plays an important role in the migration and transformation of heavy metals in rhizosphere soil and in helping plants to absorb and accumulate heavy metals or immobilize in the soil”. Don’t forget this aspect
Response 31: We sincerely thank the reviewer for careful reading. The related text has been changed as suggested.(Lines 487-489, Page 14)
Point 32: Line 502-506. rewrite
Response 32: we have rewritten this part according to the Reviewer’s suggestion.(Lines 483-487, Page 14)
Point 33: Comments on the Quality of English Language:it needs to be revised
Response 33: We apologize for the poor language of our original manuscript. We have tried our best to polish the language throughout the text and corrected the grammatical, styling, and typos found in our original manuscript. And we really hope that the revised manuscript has been improved significantly in readability and could be acceptable for you.

Reviewer 2 Report
Comments and Suggestions for Authors
The manuscript titled “Role of plant growth promoting rhizobacteria in the plant machinery for soil heavy metal detoxification” is a well-planned and described review, concerning an important aspect of the generally understood environmental safety. There are some gaps in the work that need to be filled:
- It would be advisable to include a table of contents in reviews to make reading the work easier
- Line 10: please explain the abbreviation PGPR, it is used for the first time at work
- Line 29-31: please specify in which region
- Line 37-38: please specify in what organisms they accumulate
- Line 41: please just mention an example of traditional physical and chemical treatments
- Line 42: I don't understand what the authors mean by saying soil and ecological environment. I suggest replacing it with a natural environment
- Line 51-53 and 53-55: please remove one of the sentences because they are duplicates
- Line 60-62: are the authors able to determine why there was such an increase in these years?
- Line 67-69: in figure 1b you can see that this amount does not increase, but decreases. Therefore, this sentence or Figure 1b should be slightly modified
- Line 85-87: part of the sentence (Line 87) is unnecessary
- Line 95-97: I think the abbreviation PGPR can now be used since it is explained earlier
- Line 97-101: please write the bacteria correctly, i.e. - use the species in the sentence, or use the ending spp for each name of the bacteria.
- Table 1. Please use Latin names consistently for all plants
- Line 151-152: please write the name of the bacteria correctly (as in line 97-101), please apply it to the entire manuscript
- 3.1, to the topic of atmospheric nitrogen fixation, it is also necessary to add information about the nitrogenase enzyme complex and the genes related to it, which are present in bacteria and enable nitrogen fixation.
- Line 195: please explain what the abbreviation IAA is
- Line 226-229: please rephrase the sentence because it is not understandable
- Line 507-519: this fragment is very important, I think it deserves an additional subsection in section 4, explaining it in more detail, mainly about the impact of various environmental conditions on the survival of the microorganisms being treated and their functioning as PGPR.
- I think it is worth adding a chapter/subchapter at the end of the work in which the authors briefly explain the fate of plants with accumulated heavy metaorganisms, whether heavy metals are transformed completely, whether some of the plants are utilized and if so, how
Author Response
Response to Reviewer 2 Comments
Comments and Suggestions for Authors:
The manuscript titled “Role of plant growth promoting rhizobacteria in the plant machinery for soil heavy metal detoxification” is a well-planned and described review, concerning an important aspect of the generally understood environmental safety.
Response: We thank the reviewer for the positive and constructive comments on the manuscript, which has helped to improve the quality of our papers. We tried our best to improve the manuscript and made some changes to the manuscript. These changes will not influence the content and framework of the paper. And here we did not list the changes but marked in red in the revised paper. We appreciate for Editors and Reviewers warm work earnestly and hope that the correction will meet with approval.
There are some gaps in the work that need to be filled:
Point 1: It would be advisable to include a table of contents in reviews to make reading the work easier
Response 1: We thank the reviewer for the suggestion. The graphical abstract has been added in the revised manuscript. (Page 1)
Point 2: Line 10: please explain the abbreviation PGPR, it is used for the first time at work
Response 2: We thank the reviewer for pointing this out. “PGPR” has been changed to “Plant growth-promoting rhizobacteria (PGPR)” in the revision (Line 10, Page 1). We have checked the entire text and addressed similar issues.
Point 3: Line 29-31: please specify in which region
Response 3: We thank the reviewer for pointing this out. In the revised manuscript, we have added the information about region. The text has been modified as follows: “According to the China National Soil Pollution Survey Bulletin, heavy metals Cd, Hg, As, Cu, Pb, Cr, Zn and Ni in soil exceeded the standard by 7.0%, 1.6%, 2.7%, 2.1%, 1.5%, 1.1%, 0.9% and 4.8%, respectively. ” (Lines 32-34, Page 2)
Point 4: - Line 37-38: please specify in what organisms they accumulate
Response 4: We sincerely thank the reviewer for careful reading. We explained this accumulation process of heavy metals in detail in the revision: “Because of its remarkable environmental persistence, heavy metals in soil will be ab-sorbed and accumulate in plants, then pose hazards to human and animal via food chains” (Lines 41-42, Page 2)
Point 5: Line 41: please just mention an example of traditional physical and chemical treatments
Response 5: We have modified the text as suggested: “Although traditional physicochemical treatments, such as soil replacement, soil washing and chemical solidification, partly reduced the migration of soil heavy metals to groundwater, their applications are limited due to high energy consumption, sec-ondary pollution, and the breakdown of soil aggregate structure ” (Lines 44-45, Page 2 )
Point 6: Line 42: I don't understand what the authors mean by saying soil and ecological environment. I suggest replacing it with a natural environment
Response 6: We are sorry for this confusing description. We have changed this sentence to “Although traditional physicochemical treatments such as soil replacement, soil washing and soil solidification partly reduced the migration of heavy metals in soil and to groundwater...” (Lines 45-46, Page 2)
Point 7: Line 51-53 and 53-55: please remove one of the sentences because they are duplicates
Response 7: We thank the reviewer for pointing out this mistake. The duplicate part has been deleted.
Point 8: Line 60-62: are the authors able to determine why there was such an increase in these years?
Response 8: We thank the reviewer for the comment. By reviewing the literature, we analyzed the reasons for the increase in recent years: the industrialization of heavy metal production has increased substantially, but it also brings the risk of heavy metal pollution, and the frequent pollution accidents in recent years, so there are more and more studies on the control of heavy metal pollution.
We added the explanation in the revised manuscript as: “Such sharp growth is attributed to the global environmental concern on heavy metal pollution along with the rapid development of industry.” (Lines 62-64, Page 2)
Point 9: Line 67-69: in figure 1b you can see that this amount does not increase, but decreases. Therefore, this sentence or Figure 1b should be slightly modified
Response 9: Many thanks to the reviewer for pointing this out. Since the statistical time of this table is May 2023, the number of articles on "PGPR, heavy metal" in 2023 is not complete. We have re-counted and drawn as recommended, with the deadline ending December 31,2023. In the revised manuscript, we have replaced Figure 1b (Line 65, Page 2) with the updated version.
Point 10: Line 85-87: part of the sentence (Line 87) is unnecessary
Response 10: We have deleted the sentence as suggested. (Lines 97-99, Page 2)
Point 11: Line 95-97: I think the abbreviation PGPR can now be used since it is explained earlier
Response 11: We have modified "Growth-promoting bacteria" to "PGPR" as recommended. (Line 89, Page 2)
Point 12: Line 97-101: please write the bacteria correctly, i.e. - use the species in the sentence, or use the ending spp for each name of the bacteria.
Response 12: We thank the reviewer for the comment. Based on this advice, we have checked the entire text and added the ending “spp.” for each name of the bacteria. (Lines 127-128, Page 5; Lines 184-186, Page 7; Lines 251-253, Page 8; Lines 339-340, Page 10;)
Point 13: Table 1. Please use Latin names consistently for all plants
Response 13: Thanks to the reviewer’s suggestion. we have carefully revised the Table 1 (Line 101, Pages 3-4) by unifying Latin names for all plants.
Point 14: Line 151-152: please write the name of the bacteria correctly (as in line 97-101), please apply it to the entire manuscript
Response 14: We have modified the names of the bacteria throughout the manuscript as suggested. (Lines 127-128, Page 5; Lines 184-186, Page 7; Lines 251-253, Page 8; Lines 339-340, Page 10)
Point 15: 3.1, to the topic of atmospheric nitrogen fixation, it is also necessary to add information about the nitrogenase enzyme complex and the genes related to it, which are present in bacteria and enable nitrogen fixation.
Response 15: Thank you very much for the valuable suggestions. We have incorporated and elaborated on the discussion regarding nitrogenase and nitrogen fixation genes in the manuscript. We also have added a new reference [51](Plant and Soil, 2017, 415, 245-255) and [53](Plant Stress, 2022, 4, 100080) to help us to explain them. (Lines 129-145, Pages 5-6)
Point 16: Line 195: please explain what the abbreviation IAA is
Response 16: We thank the reviewer for the comment. “IAA” is “indole-3-acetic acid (IAA)”, which has been defined in Table 1 (Line 101, Page 3). And its abbreviation IAA was used in the following next.
Point 17: Line 226-229: please rephrase the sentence because it is not understandable
Response 17: We are sorry for this confusing description.
We have revised the whole paragraph and described in a more detailed and orderly way the protection mechanism of siderophore: on the one hand, to enhance the absorption of iron ions by plants, on the other hand, to prevent excessive iron ions from producing OH· damage cells. (Lines 217-224 Page 8)
Point 18: Line 507-519: this fragment is very important, I think it deserves an additional subsection in section 4, explaining it in more detail, mainly about the impact of various environmental conditions on the survival of the microorganisms being treated and their functioning as PGPR.
Response 18: We sincerely thank the reviewer for making these valuable comments. The survival rate of PGPR is indeed an important aspect that should be considered. Because PGPR has an induced system resistance (section 3.4.3), it can function rapidly in a harsh environment. Therefore, we mainly focus on the survival rate of PGPR. Now, most studies have isolated strains from contaminated sites and introduced them into similar environments, so that PGPR can play an effective role. This makes the application of PGPR limited by the environment and time, so the development of high efficiency strains needs attention in the future.
According to the suggestion, we expanded the first section of outlook (Lines 493-504, Pages 14) to explain it in detail and added [169](Environmental Modelling & Software, 2006, 21, 8, 1158-1171) to support it.
Point 19: I think it is worth adding a chapter/subchapter at the end of the work in which the authors briefly explain the fate of plants with accumulated heavy metaorganisms, whether heavy metals are transformed completely, whether some of the plants are utilized and if so, how
Response 19: We sincerely appreciate the valuable comments.
We have checked the literature carefully but found that there is little literature on the fate of the plants after phytoremediation, most of which focus on the restored soil. Therefore, after careful consideration, we decided to take the proposed content of the reviewer as the fourth section in outlook. We also have added a new reference [170](Chemosphere, 2020, 251, 126310) to help us to explain it. (Lines 521-530, Page 15)
